# The association between multimorbidity and osteoporosis investigation and treatment in high-risk fracture patients in Australia: A prospective cohort study

Dana Bliuc[1,2]*, Thach Tran[1,3], Weiwen Chen[1], Dunia Alarkawi[1,3], Dima A. Alajlouni[1,3], Fiona Blyth[4], Lyn March[5], Kristine E. Ensrud[6,7,8], Robert D. Blank[1], Jacqueline R. Center[1,3]

1 Skeletal Diseases Program, Garvan Institute of Medical Research, Sydney, New South Wales, Australia, 2 School of Population Health, Faculty of Medicine and Health, UNSW, Sydney, New South Wales, Australia, 3 St Vincent's Clinical School, Faculty of Medicine and Health, UNSW, New South Wales, Australia, 4 Concord Clinical School, University of Sydney, Sydney, Australia, 5 Institute of Bone & Joint Research, University of Sydney, Sydney, Australia, 6 Center for Care Delivery and Outcomes Research, Minneapolis VA Healthcare System, Minneapolis, Minnesota, United States of America, 7 Department of Medicine, University of Minnesota, Minneapolis, Minnesota, United States of America, 8 Division of Epidemiology and Community Health, University of Minnesota, Minneapolis, Minnesota, United States of America

* d.bliuc@garvan.org.au

## Abstract

### Background

Multimorbidity is common among fracture patients. However, its association with osteoporosis investigation and treatment to prevent future fractures is unclear. This limited knowledge impedes optimal patient care.

This study investigated the association between multimorbidity and osteoporosis investigation and treatment in persons at high risk following an osteoporotic fracture.

### Methods and findings

The Sax Institute's 45 and Up Study is a prospective population-based cohort of 267,153 people in New South Wales, Australia, recruited between 2005 and 2009. This analysis followed up participants until 2017 for a median of 6 years (IQR: 4 to 8). Questionnaire data were linked to hospital admissions (Admitted Patients Data Collection (APDC)), emergency presentations (Emergency Department Data Collection (EDDC)), Pharmaceutical Benefits Scheme (PBS), and Medicare Benefits Schedule (MBS). Data were linked by the Centre for Health Record Linkage and stored in a secured computing environment. Fractures were identified from APDC and EDDC, Charlson Comorbidity Index (CCI) from APDC, Dual-energy X-ray absorptiometry (DXA) investigation from MBS, and osteoporosis treatment from PBS.

Out of 25,280 persons with index fracture, 10,540 were classified as high-risk based on 10-year Garvan Fracture Risk (age, sex, weight, prior fracture and falls) threshold ≥20%. The association of CCI with likelihood of investigation and treatment initiation was

**Data Availability Statement:** Data cannot be shared publicly as they are owned by the Sax Institute, which is the data custodian for the 45 and

Up Study. Anyone wishing to request access can do so through the 45 and Up Study. The data are available for researchers who meet the criteria for access to confidential data. Details are available at https://www.saxinstitute.org.au/our-work/45-up-study/ and requests to access data can be submitted via email to 45andUp.Research@saxinstitute.org.au.

**Funding:** This work was supported by an Amgen Foundation (Award number: Competitive Bone Grant ProliaBCGP-06 to DB), a National Health & Medical Research Council Grant (Award number: 1108886 to JRC), a Medical Research Future Fund (Award number: 1137462 to JRC) and Mrs Gibson and Ernst Heine Family Foundation (Award number: none). The funders had no role in study design, data collection and analysis, decision to publish, or preparation of the manuscript.

**Competing interests:** I have read the journal's policy and the authors of this manuscript have the following competing interests: WC has given educational talks for Amgen. KEE has received grants from the National Institute of Health. RDB has been a consultant for Bristol Myers Squibb, served on an advisory board for Amgen, received authorship royalties from Wolters Kluwer, received an editorial stipend from Elsevier, received travel support from Amgen, and owns stock in Abbott Labs, Abbvie, Amgen, JangoBio, and Procter & Gamble. JRC has been on a medical advisory board for Amgen and given educational talks for Amgen and Teva. DB, TT, LM, FB, DA and DA have no conflict of interest to declare.

**Abbreviations:** ACHI, Australian Classification of Health Interventions; APDC, Admitted Patients Data Collection; BMD, bone mineral density; CCI, Charlson Comorbidity Index; CheReL, Centre for Health Record Linkage; DXA, Dual-energy X-ray absorptiometry; EDDC, Emergency Department Data Collection; MBS, Medicare Benefits Schedule; PBS, Pharmaceutical Benefits Scheme; SERMS, selective estrogen receptor modulators; SNOMED-CT, Systematised Nomenclature of Medicine-Clinical Terms.

determined by logistic regression adjusted for education, socioeconomic and lifestyle factors). The high-risk females and males averaged 77 ± 10 and 86 ± 5 years, respectively; >40% had a CCI ≥2. Only 17% of females and 7% of males received a DXA referral, and 22% of females and 14% males received osteoporosis medication following fracture. A higher CCI was associated with a lower probability of being investigated [adjusted OR, females: 0.73 (95% CI, 0.61 to 0.87) and 0.43 (95% CI, 0.30 to 0.62); males: 0.47 (95% CI, 0.33 to 0.68) and 0.52 (0.31 to 0.85) for CCI: 2 to 3, and ≥4 versus 0 to 1, respectively] and of receiving osteoporosis medication [adjusted OR, females: 0.85 (95% CI, 0.74 to 0.98) and 0.78 (95% CI, 0.61 to 0.99); males: 0.75 (95% CI, 0.59 to 0.94) and 0.37 (95% CI, 0.23 to 0.53) for CCI: 2 to 3, and ≥4 versus 0 to 1, respectively]. The cohort is relatively healthy; therefore, the impact of multimorbidity on osteoporosis management may have been underestimated.

## Conclusions

Multimorbidity contributed significantly to osteoporosis treatment gap. This suggests that fracture risk is either underestimated or underprioritized in the context of multimorbidity and highlights the need for extra vigilance and improved fracture care in this setting.

## Author summary

### Why was this study done?

- Osteoporotic fractures are common, costly, and associated with increased risk of future fracture and premature mortality.

- Effective preventive treatment is available and recommended for high-risk patients with a 10-year fracture risk >20%, but its uptake in this group is suboptimal.

- High-risk patients often have multiple chronic conditions, and it is important to know to what degree the other chronic conditions might alter the uptake of fracture prevention medication in order to optimise patient care.

### What did the researchers do and find?

- We conducted a prospective study including over 10,000 adults aged 50+ with an osteoporotic fracture and high-risk of future fracture to investigate the association between multimorbidity and osteoporosis investigation and treatment.

- In this high-risk group, the vast majority of adults were not investigated or treated for osteoporosis following an osteoporotic fracture.

- Adults with multiple chronic conditions were significantly less likely to be investigated [adjusted OR, females: 0.73 (95% CI, 0.61 to 0.87) and 0.43 (95% CI, 0.30 to 0.62); males: 0.47 (95% CI, 0.33 to 0.68) and 0.52 (0.31 to 0.85) for CCI: 2 to 3, and ≥4 versus 0 to 1, respectively] or treated for osteoporosis [adjusted OR, females: 0.85 (95% CI, 0.74 to 0.98) and 0.78 (95% CI, 0.61 to 0.99); males: 0.75 (95% CI, 0.59 to 0.94) and 0.37 (95%

CI, 0.23 to 0.53) for CCI: 2 to 3, and ≥4 versus 0 to 1, respectively] compared to those without chronic conditions.

### What do these findings mean?

- The presence of additional chronic conditions was associated with a lower likelihood of being investigated and treated for osteoporosis.

- These findings suggest that fracture preventing treatment is either underprioritised or underestimated in the presence of other chronic conditions.

- There is a need for more awareness of increased fracture risk in complex patients with multiple chronic conditions.

## Introduction

One in 3 females and 1 in 5 males over the age of 50 [1] suffer osteoporotic fractures, resulting in disability, loss of independence [2], subsequent fracture [3], and premature mortality [4,5]. Fractures in older adults result in significant financial cost and diversion of resources to aged care. This burden will increase as the population ages [6].

Antifracture medication reduces risk of subsequent fracture [7] and may prevent premature mortality [8,9]. Thus, drug treatment is advised following hip and clinical vertebral fracture [10–12] and for other fractures if bone density is low. Some guidelines also recommend drug treatment for a 10-year fracture risk estimate ≥20% or ≥40% [13]. Despite these recommendations, fracture prevention is suboptimal worldwide [14–17], with <30% of females and <20% of males treated following an osteoporotic fracture. Furthermore, clinical practice patterns have not improved over time [16,18].

Osteoporosis is commonly accompanied by other chronic conditions. A recent study reported that 2/3 of individuals with osteoporosis had more than two other conditions [19]. However, there is a lack of evidence of treatment efficacy in complex patients, as the major trials of fracture prevention drugs were conducted in relatively healthy individuals without major comorbidities. Furthermore, the role of major comorbidities on fracture prevention management is unknown. We hypothesised that multimorbidity, defined as number and severity of chronic conditions present at the time of osteoporotic fracture, decreases the rate of osteoporosis investigation and treatment. We studied this in people with an index minimal trauma fracture and high risk of subsequent fracture participating in a well-characterized large population-based study.

## Methods

This is a prospective study involving a large population-based study involving questionnaire data linked to administrative health databases. This study was developed as a part of a successful funding proposal (2020 Amgen International Competitive Bone Grant Prolia BCGP-06) to determine the magnitude of association between multimorbidity and osteoporosis treatment gap in patients at high risk of fracture. The definition for each of the high-risk groups, multimorbidity and treatment groups were predefined prior to the commencement of data analysis,

as were the details of the prevented fracture estimation. The sensitivity analyses and the risk of subsequent fracture and mortality were added during revision process.

This study is reported as per the Strengthening the Reporting of Observational Studies in Epidemiology (STROBE) guideline (S1 STROBE Checklist).

## Participants and setting

This study included females and males aged 45 and over enrolled in the Sax Institute 45 and Up Study with a confirmed index fracture. The 45 and Up Study is an ongoing prospective cohort study that aims to characterise population health and ageing in New South Wales, Australia, through detailed questionnaires and linkage of all available administrative health databases. A detailed description of the study design and recruitment has been published previously [20,21]. Participants were randomly sampled from the Services Australia (formerly the Australian Government Department of Human Services), Medicare enrolment database, which provides near complete coverage of the population. The target population comprised females and males over 45 years of age with oversampling of the rural population and individuals aged over 80, to account for death and loss of follow-up. Recruitment took place between 2005 and 2009 and the response rate was 18%.

## Study design, data collection, and management

### Data collection

Participants completed baseline and 5-year follow-up questionnaires and consented to have their medical records linked to administrative health databases.

**Questionnaires.** Demographic data (age, sex, weight, date of recruitment), lifestyle factors (physical activity, smoking), comorbidities, residency, and disability, and socioeconomic factors (education, private health insurance) were collected using a standard questionnaire at baseline and again in a second wave approximately 5 years following recruitment.

### Linked administrative databases

All participants had their medical records linked to NSW Ministry of Health Emergency Department Data Collection (EDDC), NSW Ministry of Health Admitted Patient Data Collection (APDC), NSW Registry of Births, Deaths, and Marriages (RBDM), Pharmaceutical Benefit Scheme (PBS), and Medicare Benefits Schedule (MBS).

EDDC contains information on emergency presentations and APDC on hospital admission to all NSW hospitals and private day procedure centres. RBDM contain information on death dates and cause. PBS contains information on drug prescriptions and MBS on medical attendances through Medicare claims.

Probabilistic record linkage of all medical records (questionnaire and administrative datasets) was performed by the NSW Centre for Health Record Linkage (CHeReL). CHeReL uses a probabilistic procedure to link records, in which records with an uncertain probability of being true matches are checked by hand. Its current estimated false positive rate is 0.5% (http://www.cherel.org.au). The linkage to MBS and PBS data was deterministic and data were supplied to the Sax Institute by Services Australia. The Sax Institute's 45 and Up Study was approved by UNSW Human Research Ethics Committee. Ethics approval for the current study was obtained from the NSW Population Health Services Research Ethics Committee. All participants provided signed consent to take part in the study and to have their data linked to administrative datasets.

## Cohort selection

This analysis was based on participants who experienced a minimal trauma fracture during the study follow-up and classified as "high-risk" for future fracture.

**Fracture identification.** Fractures were identified using International Classification of Diseases (ICD-10) and Systematised Nomenclature of Medicine-Clinical Terms (SNO-MED-CT) codes from the ADDC and EDDC datasets in conjunction with the Australian Classification of Health Interventions (ACHI) procedure codes as previously reported [4]. High trauma fractures as well as fractures of digits, skull, and face were excluded from the analysis as they are commonly considered "non-osteoporotic." High trauma fractures were identified from ICD-10 codes. In addition, the occurrence of more than 4 fractures of different sites during the same event, was likely the result of high trauma and were excluded.

Fracture events were classified according to the recruitment date as prior fractures or incident fractures. Index fracture represented the first fracture after the recruitment date. Index fracture was classified according to site: hip, clinical vertebral, proximal (all fractures above the elbow and above the knee), and distal (remaining fracture sites). If more than one fracture occurred in the same event, the most proximal site was chosen.

**Definition of high-risk group and participant selection.** Ten-year fracture risk was calculated using the Garvan Fracture Risk calculator, based on age, sex, weight, prior fractures (including the index fractures), and falls within the last 12 months. Weight and number of falls were obtained from the questionnaire closest to the time of fracture.

Participants with an index fracture, a 10-year Garvan risk $\geq$20%, and not receiving fracture-preventing medications at the time of the index fracture were included (Fig 1).

## Multimorbidity definition and identification

Multimorbidity was determined using Charlson Comorbidity Index (CCI). In addition, individual comorbidities either included in CCI or self-reported in questionnaire (i.e., hypertension, rheumatoid arthritis, Parkinson disease, anxiety, depression) were also identified.

CCI was estimated from hospital admissions dataset (APDC) and was based on ICD-10 codes using a 5-year look back window prior to index fracture. Chronic conditions included in CCI were identified from all 50 hospital diagnosis codes available. Previous studies reported high positive predictive value for CCI obtained from administrative registries [22]. A full list of the codes or self-report diagnoses used to identify chronic conditions is provided in the Supporting information (S1 Table). CCI was classified in 3 groups: low morbidity CCI $\leq$1 (referent group), mild morbidity 2 to 3, and severe morbidity $\geq$4. These groups were chosen to discriminate between those with small comorbidity burden (up to 1 chronic condition), mild morbidity (2 to 3 chronic conditions), and severe morbidity (more than 4 chronic conditions).

Multimorbidity was also assessed as number of comorbidities, and number of hospitalisations within 5 years prior to index fracture.

In addition to CCI and the unweighted count of conditions, specific conditions with a prevalence $\geq$3% were analysed and reported separately. These specific chronic conditions were identified using three data sources: self-reported, hospital admissions, and prescription databases cumulatively for up to 5 years prior to index fracture. In the questionnaires, participants were asked whether "a doctor has ever told you that you have a chronic condition (yes/no)." The ATC codes in the prescription database (PBS) were used to derive information about chronic conditions using the Rx-Risk Index [22,23]. The Rx-Risk model mapped up to 46 conditions using ATC medication codes from PBS (S1 Table).

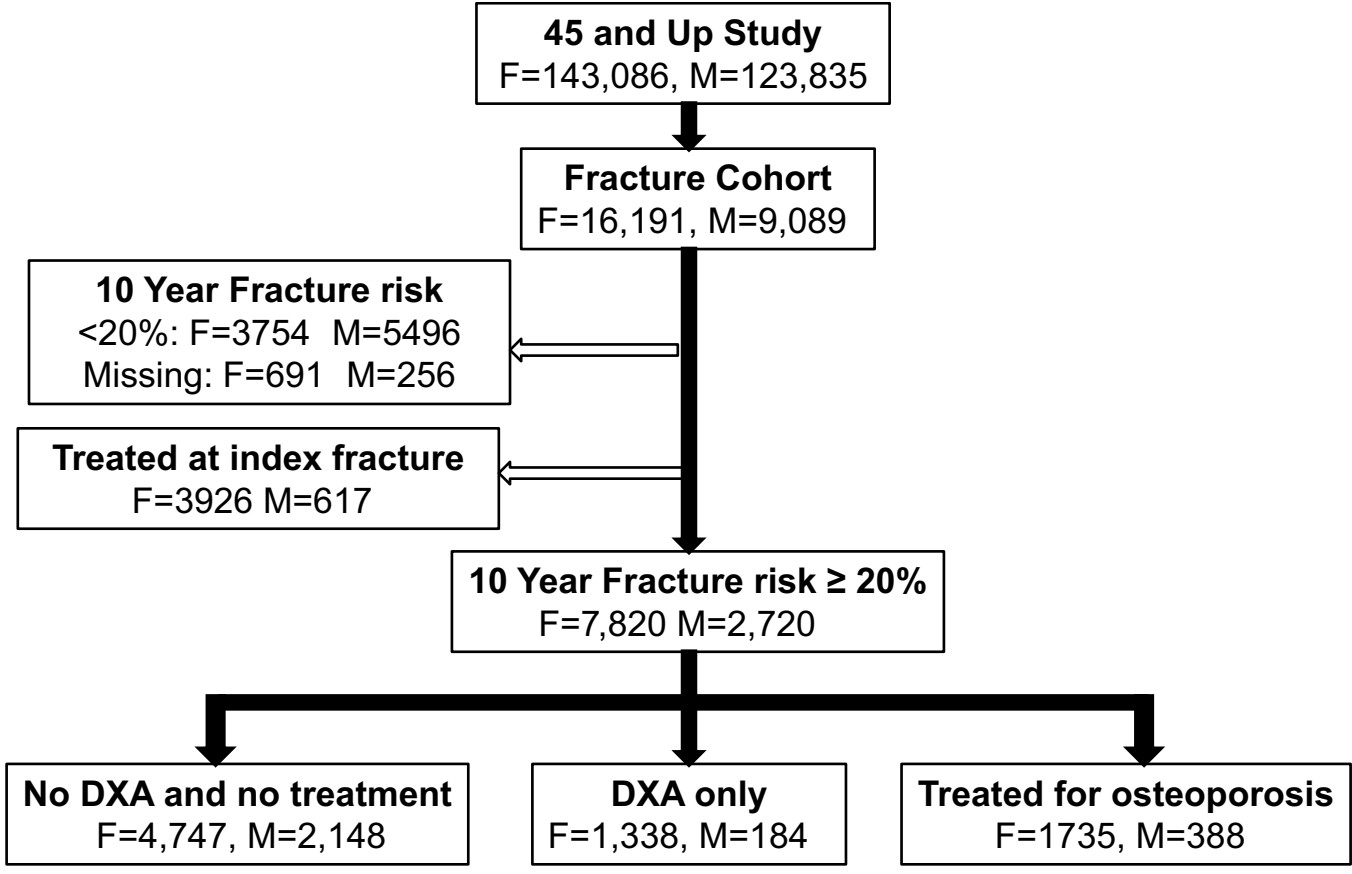

**Fig 1. Flow chart and sample selection.**

### Risk factors

The risk factors analysed in this study were age, education, marital status, place of residency, private health insurance, and lifestyle factors (smoking and alcohol consumption).

Previous studies have shown that the risk of fracture as well as the likelihood of osteoporosis treatment is significantly higher for females [24]. Therefore, all the models were stratified by sex.

The study cohort included participants who reported information on weight and all the risk factors listed above.

**Primary outcome measurements.** The primary outcome measurements included investigation for osteoporosis by DXA and treatment for osteoporosis initiated after the index fracture. Both were identified from nationwide administrative databases, ensuring a complete ascertainment for all participants.

**Investigation for osteoporosis.** Osteoporosis investigation was based on DXA referrals identified from MBS. All DXA referrals were extracted for each individual for the entire study follow-up (from enrolment until April 2017) and classified as prior or new based on the index fracture date. The median follow-up for new DXA was 3 years (IQR 1 to 5) following index fracture. Prior DXA was used as a covariate in the model of new DXA prediction.

**Osteoporosis treatment.** Osteoporosis pharmacotherapy included bisphosphonates, denosumab, teriparatide, strontium ranelate, selective estrogen receptor modulators (SERMS), and hormone therapy. Medication was identified from PBS data set (S1 Table). There was no restriction based on the number of prescriptions filled.

Participants were classified as prior users, new users, and never users. Prior users were classified as any medication initiated within 5 years prior to the index fracture and excluded. New users included all participants who received at least one prescription for any of the osteoporosis medications during a median of 3 years (IQR: 1 to 5) following index fracture. Participants who had never been prescribed any drug therapy for osteoporosis during the entire follow-up were classified as never users.

## Statistical analysis

All statistical analyses were performed using SAS 9.4 and R statistical environment on a Windows platform.

Baseline characteristics were compared according to the main outcome measurements using Student $t$ tests or $\chi^2$ tests as appropriate. Given the differences in fracture risk between females and males, all analyses conducted in this study were sex specific. However, given that it is well known that males are less likely to be investigated and treated than females [24], we also performed an exploratory analysis to determine whether the association of multimorbidity with outcomes was significantly different between sexes using an interaction term between sex and CCI.

## The association of multimorbidity with DXA investigation

Given that physicians can start osteoporosis treatment based on prior DXA or other clinical considerations such as recent fracture, this analysis was limited to participants who did not receive any osteoporosis treatment after the index fracture (Fig 1).

The association of multimorbidity with subsequent odds of DXA investigation was assessed using sex-specific logistic regression models further adjusted for age, prior DXA referrals, lifestyle, private health insurance, socioeconomic factors, and education. The risk factors included in the multivariable models were chosen due to their association with fracture risk (age, lifestyle factors), impact on outcome (prior DXA) or "healthy user" measure (lifestyle factors, socioeconomic factors, and education).

## The association of multimorbidity with treatment initiation

The relationship between treatment initiation and multimorbidity was assessed using sex-specific logistic regression models further adjusted as for the DXA model.

## Estimation of the number of fractures prevented if cohort was treated

We performed an exploratory analysis to determine the number of fractures potentially prevented by osteoporosis treatment. Based on findings of a meta-analysis evaluating effect of bisphosphonate treatment on fracture risk, we assumed that treatment would reduce fracture risk by 30% [23–25].

The follow-up time was calculated from the date of index fracture to the date of second any/hip fracture, or death or end of follow-up (April 2017), whichever came first. The observed rate of any second and hip second fracture was calculated as number of events over 1,000 person-years follow-up, assuming a Poisson distribution. The number of prevented fractures was

then obtained by subtracting from the observed number the estimated number of any/hip subsequent fracture occurring with an incidence rate lower by 30% than the observed rate.

Given that multimorbidity is also associated with increased risk of mortality, we have also performed a cause-specific competing risk model for subsequent fracture, accounting for competing risk of death [26,27].

## Sensitivity analyses

We have performed three sensitivity analysis including (1) group with 10-year Garvan Fracture Risk ≥40%; (2) all fractures, as well as hip and vertebral fractures regardless of the 10-year Garvan Fracture Risk; and (3) investigation by DXA in all participants regardless of whether they received osteoporosis treatment.

The first sensitivity analysis was performed because clinical guidelines in some countries recommend osteoporosis treatment at a higher fracture risk threshold. The second sensitivity analysis, requested by a reviewer, was performed because most clinical guidelines recommend osteoporosis investigation and treatment initiation following all hip and vertebral fracture. Given that comorbidities are associated with factors included in the Garvan Fracture Risk calculator, we conducted an additional analysis in all fracture group.

The third sensitivity analysis was performed as requested to determine whether the exclusion of the treated participants affected the main outcomes.

## Results

### Cohort characteristics

Of the total 267,153 participants (266,921 with accurate records) enrolled in the 45 and Up cohort, 25,280 (16,191 females and 9,089 males) sustained an index osteoporotic fracture. Of the total individuals with index fracture, 15,083 (60%) were classified as being at high risk of subsequent fracture based on a 10-year Garvan Fracture Risk Calculator estimate ≥20% (Fig 1). More females (76%) than males (61%) were classified into the high-risk group.

Compared to the low-risk group, in the high-risk group, individuals were older [age, years mean (SD), females: 77 (10) versus 57 (4) and males: 86 (5) versus 65 (8), for high- versus low-risk groups, respectively] and had more severe fractures [index hip and vertebral fracture n (%), females: 3,051 (26%) versus 223 (6%) and males: 1,500 (45%) versus 981 (18%) for high-versus low-risk groups, respectively]. Those in the high-risk group also had a higher CCI [females, (≤1, 2 and 3, ≥4) was 70%, 24%, 7% for high-risk and 88%, 9%, 3% for low-risk ($p$-value < 0.001); males 47%, 38%, 15% for high, and 75%, 18%, 7% for low-risk ($p$-value < 0.001)] and a higher prevalence of cardiovascular, diabetes, respiratory, renal disease, cancer, and dementia. There was no difference in the level of education between the two fracture risk groups.

A further 4,543 individuals were already on osteoporosis treatment and excluded from analysis.

The analytical cohort consisted of 10,540 females and males with a 10-year risk of subsequent fracture ≥20% who were not on treatment at the time of index fracture. Females averaged 77 (10) years of age, had a 26% prevalence of index fracture at the hip or spine, and 30% prevalence of CCI ≥2. Males averaged 86 (5) years of age, had 45% prevalence of index fracture at the hip or spine, and 53% prevalence of CCI ≥2 (Table 1).

Of the 7,820 females, 1,338 (17%) received only DXA referral, and a further 1,735 (22%) received osteoporosis medication during a median follow-up of 3 years (IQR: 1 to 5). Of the 2,720 males, 184 (7%) received only DXA referral, 388 (14%) received osteoporosis medication during the same follow-up time (Fig 1).

**Table 1. Characteristics at the time of index fracture for females and males in the high-risk group according to DXA referral and treatment initiation.**

| | Females | | | | | | | | Males | | | | | | | |
| --- | --- | --- | --- | --- | --- | --- | --- | --- | --- | --- | --- | --- | --- | --- | --- | --- |
| | DXA | | No DXA | | Treatment initiation[1] | | No treatment initiation | | DXA | | No DXA | | Treatment initiation[1] | | No treatment initiation | |
| | N | (%/SD) | N | (%/SD) | N | (%/SD) | N | (%/SD) | N | (%/SD) | N | (%/SD) | N | (%/SD) | N | (%/SD) |
| Number | 1,338 | | 4,747 | | 1,735 | | 6,085 | | 184 | | 2,148 | | 388 | | 2,332 | |
| Age, years[2] | 71 | (8) | 77 | (11) | 76 | (9) | 75 | (10) | 84 | (4) | 87 | (5) | 85 | (4) | 87 | (5) |
| Fracture site | | | | | | | | | | | | | | | | |
| Hip | 107 | (8) | 890 | (19) | 411 | (24) | 997 | (16) | 33 | (18) | 752 | (35) | 163 | (42) | 785 | (34) |
| Vertebral | 48 | (4) | 268 | (6) | 98 | (6) | 316 | (5) | 21 | (11) | 231 | (11) | 44 | (11) | 252 | (11) |
| Proximal | 297 | (22) | 1,372 | (29) | 439 | (25) | 1,669 | (27) | 74 | (40) | 827 | (39) | 115 | (30) | 901 | (39) |
| Distal | 886 | (66) | 2,217 | (47) | 787 | (45) | 3,103 | (51) | 56 | 30) | 338 | 16) | 66 | 17) | 394 | 17) |
| Number of comorbidities | | | | | | | | | | | | | | | | |
| 0 | 234 | (17) | 746 | (16) | 293 | (17) | 980 | (16) | 6 | (3) | 134 | (6) | 32 | (8) | 140 | (6) |
| 1–2 | 617 | (46) | 2,077 | (44) | 745 | (43) | 2,694 | (44) | 84 | (46) | 851 | (40) | 176 | (45) | 935 | (40) |
| ≥3 | 487 | (36) | 1,924 | (41) | 697 | (40) | 2,411 | (40) | 94 | (51) | 1,163 | (54) | 180 | (46) | 1,257 | (54) |
| Number of prior hospitalisations | | | | | | | | | | | | | | | | |
| 0 | 1,080 | (81) | 3,323 | (70) | 1,281 | (74) | 4,403 | (72) | 99 | (54) | 935 | (44) | 219 | (56) | 1,034 | (44) |
| 1–2 | 236 | (18 | 1,192 | (25) | 402 | (23) | 1,428 | (23) | 69 | (38) | 904 | (42) | 142 | (37) | 973 | (42) |
| ≥3 | 22 | () | 232 | (5) | 52 | (3) | 254 | (4) | 16 | (9) | 309 | (14) | 27 | (7) | 325 | (14) |
| Charlson index | | | | | | | | | | | | | | | | |
| ≤1 | 1,103 | (82) | 3,283 | (69) | 1,295 | (75) | 4,386 | (72) | 116 | (63) | 951 | (44) | 221 | (57) | 1,067 | (46) |
| 2–3 | 197 | (15) | 1,110 | (23) | 347 | (20) | 1,307 | (21) | 47 | (26) | 837 | (39) | 139 | (36) | 884 | (38) |
| ≥4 | 38 | (3) | 354 | (7) | 93 | (5) | 392 | (6) | 21 | (11) | 360 | (16) | 28 | (7) | 381 | (16) |
| Hypertension | 680 | (51) | 2,559 | (54) | 908 | (52) | 3,239 | (53) | 111 | (60) | 1,207 | (56) | 218 | (56) | 1,318 | (57) |
| Acute myocardial infarction | 56 | (4) | 314 | (7) | 93 | (5) | 370 | (6) | 26 | (14) | 301 | (14) | 50 | (13) | 327 | (14) |
| Congestive heart disease | 191 | (14) | 857 | (18) | 305 | (18) | 1,048 | (17) | 75 | (41) | 747 | (35) | 120 | (31) | 822 | (35) |
| Stroke | 39 | (3) | 238 | (5) | 77 | (4) | 277 | (5) | 14 | (8) | 234 | (11) | 27 | (7) | 248 | (11) |
| Diabetes | 34 | (3) | 215 | (5) | 35 | (2) | 249 | (4) | 8 | (4) | 124 | (6) | 11 | (3) | 132 | (6) |
| Peripheral vascular disease | * | * | 35 | (0.2% | 13 | (1) | 38 | (1) | * | * | 37 | (2) | * | * | 39 | (2) |
| Chronic pulmonary disease | 275 | (21) | 877 | (18) | 310 | (18) | 1,152 | (19) | 32 | (17) | 343 | (16) | 59 | (15) | 375 | (16) |
| Renal disease | 13 | (1) | 129 | (3) | 30 | (2) | 142 | (2) | 10 | (5) | 147 | (7) | 14 | (4) | 157 | (7) |
| Mild liver disease | * | * | 24 | (0.5% | * | * | 29 | (0.5) | * | * | 6 | (0.3) | * | * | 6 | (0) |
| Severe liver disease | 6 | (0.5) | 31 | (0.7) | 6 | (0.3) | 37 | (0.6) | 8 | (0.5) | 16 | (0.7) | * | * | 17 | (1) |
| Cancer | 265 | (20) | 978 | (21) | 359 | (21) | 1,243 | (20) | 67 | (36) | 697 | (32) | 124 | (32) | 764 | (33) |
| Rheumatoid arthritis | 20 | (2) | 62 | (1) | 41 | (2) | 82 | (1) | * | * | 17 | (0.8) | 7 | (2) | 18 | (1) |
| Peptic ulcer | 266 | (20) | 772 | (16) | 259 | (15) | 1,038 | (17) | 21 | (11) | 310 | 14) | 64 | 16) | 331 | 14) |
| Neurological conditions | 30 | (2) | 172 | (4) | 51 | 3) | 202 | (3) | 8 | (4) | 148 | (7) | 23 | (6) | 156 | (7) |
| Dementia | 6 | (1) | 183 | (4) | 25 | (1) | 189 | (3) | * | * | 146 | (7) | 19 | (5) | 148 | (6) |
| Parkinson disease | 14 | (1) | 57 | (1) | 32 | (2) | 71 | (1) | * | * | 46 | (2) | 14 | (4) | 50 | (2) |
| Depression | 313 | (24) | 1111 | (23) | 394 | (23) | 1,424 | (23) | 24 | (13) | 313 | (15) | 20 | (5) | 102 | (4) |
| Anxiety | 158 | (12) | 537 | (11) | 215 | (12) | 695 | (11) | 13 | (7) | 89 | (4) | 63 | (16) | 337 | (14) |
| Aged care residency | 23 | (2) | 171 | (4) | 39 | (2) | 194 | (3) | * | * | 80 | (4) | 16 | (4) | 82 | (4) |
| Disability | 79 | (6) | 506 | (11) | 182 | (10) | 585 | (10) | 8 | (4) | 232 | (11) | 46 | (12) | 240 | (10) |
| Smoker | 64 | (5) | 254 | (5) | 96 | (6) | 318 | (5) | * | * | 72 | (3) | 14 | (4) | 75 | (3) |
| Marital status | 515 | (39) | 2,211 | (47) | 772 | (44) | 2,726 | (45) | 58 | (32) | 661 | (31) | 128 | (33) | 719 | (31) |
| Private health insurance | 873 | (65) | 2,603 | (55) | 988 | (57) | 3476 | (57) | 120 | (65) | 1,017 | (47) | 207 | (53) | 1,137 | (49) |

All risk factors are significantly different (p-value < 0.05) between investigated by DXA vs no investigated and treated vs not treated for both sexes.

*Indicate cell where the frequency of the conditions was reported in fewer than 5 participants. Publication of such small numbers raises confidentiality risk and is prohibited by The Sax Institute's 45 and Up Study.

[1]Treatment included at least one prescription of bisphosphonates, denosumab, teriparatide, strontium ranelate, selective estrogen receptor modulators (SERMS), and hormone therapy initiated at any time following the index fracture.

All figures are N (%) unless otherwise stated.

[2]Mean (SD).

### The association of multimorbidity with DXA

Compared to low multimorbidity (CCI ≤1), mild multimorbidity (CCI between 2 and 3) and severe multimorbidity (CCI ≥4) were significantly associated with lower odds of DXA. The magnitude of association ranged between 27% and 53% for mild multimorbidity [adjusted OR, 0.73 (95% CI, 0.61 to 0.87) and 0.47 (95% CI, 0.33 to 0.68) for females and males, respectively] and between 48% and 57% for severe multimorbidity [adjusted OR, 0.43 (0.30 to 0.62) and 0.52 (0.31 to 0.84) for females and males, respectively]. A higher number of hospitalisations was also associated with 18% to 55% lower likelihood of DXA (Table 2). Notably, the association of CCI with DXA referral was significantly higher in males compared to females (P < 0.002 for the interaction between sex and CCI).

Specific comorbidities associated with a lower likelihood of DXA referral were dementia and diabetes, with 27% to 80% lower odds of new DXA. Other specific comorbid conditions including arrhythmias, renal disease, and stroke appeared to be associated with lower odds of DXA referral, but 95% CI around the point estimates of association overlapped 1.00.

**Table 2. Clinical risk factors associated with DXA investigation following index fracture in the high-risk group.**

| Clinical risk factors | | Females | | Males | |
|---|---|---|---|---|---|
| | | Age adjusted | Multivariable adjusted | Age adjusted | Multivariable adjusted |
| | | OR (95% CI) | OR (95% CI) | OR (95% CI) | OR (95% CI) |
| **Age** | | **0.77 (0.74–0.79)** | **0.75 (0.72–0.77)** | **0.44 (0.36–0.53)** | **0.44 (0.36–0.54)** |
| Number of comorbidities | | | | | |
| | 0 | Reference | Reference | Reference | Reference |
| | 1–2 | 1.05 (0.93–1.18) | 1.08 (0.95–1.22) | 1.09 (0.69–1.74) | 1.05 (0.65–1.70) |
| | ≥3 | 1.07 (0.95–1.21) | 1.10 (0.96–1.22) | 1.01 (0.64–1.59) | 0.96 (0.60–1.54) |
| Number of prior hospitalisations | | | | | |
| | 0 | Reference | Reference | Reference | Reference |
| | 1–2 | **0.78 (0.66–0.91)** | **0.82 (0.69–0.96)** | **0.72 (0.52–0.99)** | 0.74 (0.53–1.03) |
| | ≥3 | **0.42 (0.27–0.66)** | **0.45 (0.28–0.72)** | **0.49 (0.28–0.85)** | **0.53 (0.30–0.93)** |
| Charlson Index | | | | | |
| | ≤1 | Reference | Reference | Reference | Reference |
| | 2–3 | **0.70 (0.59–0.83)** | **0.73 (0.61–0.87)** | **0.47 (0.33–0.67)** | **0.47 (0.33–0.68)** |
| | ≥4 | **0.42 (0.29–0.59)** | **0.43 (0.30–0.62)** | **0.48 (0.30–0.78)** | **0.52 (0.31–0.85)** |
| Acute myocardial infarction | | 0.95 (0.70–1.28) | 0.99 (0.70–1.41) | 1.07 (0.69–1.67) | 1.09 (0.76–1.55) |
| Ischaemic heart disease | | **1.44 (1.18–1.76)** | **1.48 (1.20–1.82)** | **1.68 (1.11–2.52)** | 1.50 (0.98–2.29) |
| Arrhythmias | | 0.80 (0.61–1.04) | 0.80 (0.61–1.04) | 0.74 (0.48–1.13) | 0.70 (0.45–1.08) |
| Stroke | | 0.89 (0.63–1.27) | 1.05 (0.69–1.59) | 0.72 (0.41–1.27) | 0.69 (0.36–1.32) |
| Diabetes | | **0.62 (0.43–0.90)** | **0.62 (0.42–0.91)** | 0.73 (0.35–1.53) | 0.73 (0.34–1.56) |
| Respiratory disease | | 1.07 (0.92–1.25) | 1.12 (0.96–1.31) | 1.07 (0.71–1.60) | 1.10 (0.73–1.68) |
| Renal disease | | **0.52 (0.29–0.92)** | 0.55 (0.30–1.07) | 0.85 (0.44–1.67) | 0.86 (0.44–1.71) |
| Dementia | | **0.21 (0.09–0.47)** | **0.23 (0.10–0.53)** | **0.16 (0.04–0.65)** | **0.20 (0.05–0.83)** |
| Cancer | | 1.06 (0.91–1.24) | 1.06 (0.91–1.25) | 1.17 (0.85–1.61) | 1.12 (0.81–1.56) |
| Peptic ulcer | | 1.32 (1.12–1.54) | **1.33 (1.13–1.57)** | 0.70 (0.44–1.13) | 0.70 (0.43–1.15) |
| Having a prior DXA | | **1.40 (1.23–1.60)** | **1.33 (1.16–1.52)** | **1.90 (1.35–2.68)** | **1.75 (1.23–2.49)** |
| Aged care residency | | **0.61 (0.39–0.95)** | 0.67 (0.42–1.08) | 0.42 (0.10–1.76) | 0.57 (0.14–2.43) |
| Disability | | **0.67 (0.52–0.86)** | 0.78 (0.60–1.02) | **0.38 (0.18–0.79)** | 0.51 (0.24–1.07) |
| Smoking | | **0.61 (0.46–0.82)** | **0.67 (0.50–0.90)** | 0.34 (0.11–1.11) | 0.26 (0.06–1.09) |
| Private health insurance | | **1.34 (1.18–1.53)** | **1.25 (1.09–1.44)** | **1.88 (1.36–2.59)** | **1.73 (1.24–2.40)** |
| Married | | 1.11 (0.98–1.26) | 1.00 (0.87–1.14) | 1.15 (0.83–1.61) | 1.24 (0.88–1.74) |

Favourable factors included private health insurance and prior DXA.

### The association of multimorbidity with osteoporosis treatment

Similar to DXA referral, mild multimorbidity (CCI between 2 and 3) and severe multimorbidity (CCI ≥4) were associated with lower odds of treatment. The magnitude of association ranged between 15% and 25% for mild multimorbidity [adjusted OR, 0.85 (95% CI, 0.74 to 0.98) and 0.75 (95% CI, 0.59 to 0.94), for females and males, respectively] and between 22% and 63% for severe multimorbidity [adjusted OR, 0.78 (95% CI, 0.61 to 0.99) and 0.37 (95% CI, 0.23 to 0.53), for females and males, respectively]. A higher number of prior hospitalisations was also significantly associated with 31% to 63% lower odds of treatment initiation (Table 3 and Fig 2). The association of CCI and prior hospitalisations with medication initiation was significantly higher in males than females ($p < 0.001$ for the interaction with sex).

Diabetes was associated with lower odds of osteoporosis pharmacotherapy in both sexes. Dementia and peptic ulcer were significantly associated with lower odds in females, and stroke and renal disease in males (Table 3).

DXA referral increased the likelihood of treatment initiation by 7- to 11-fold in both sexes.

**Table 3. Clinical risk factors associated with osteoporosis treatment initiation following index fracture.**

| Clinical risk factors | Females | | Males | |
|---|---|---|---|---|
| | Age-adjusted OR (95% CI) | Multivariate OR (95%CI) | Age-adjusted OR (95% CI) | Multivariate OR (95% CI) |
| Age | 1.01 (0.99–1.04) | 1.02 (0.99–1.05) | 0.69 (0.61–0.77) | 0.68 (0.60–0.77) |
| Number of comorbidities | | | | |
| 0 | Reference | Reference | Reference | Reference |
| 1–2 | 0.91 (0.78–1.07) | 0.93 (0.79–1.09) | 0.80 (0.52–1.21) | 0.74 (0.48–1.14) |
| ≥3 | 0.95 (0.81–1.11) | 0.94 (0.80–1.11) | **0.60 (0.39–0.91)** | **0.53 (0.35–0.81)** |
| Number of prior hospitalisations | | | | |
| 0 | Reference | Reference | Reference | Reference |
| 1–2 | 0.95 (0.84–1.00) | **0.92 (0.80–0.99)** | **0.68 (0.54–0.86)** | **0.66 (0.52–0.84)** |
| ≥3 | **0.69 (0.51–0.94)** | **0.69 (0.50–0.94)** | **0.39 (0.25–0.59)** | **0.37 (0.24–0.57)** |
| Charlson index | | | | |
| ≤1 | Reference | Reference | Reference | Reference |
| 2–3 | **0.86 (0.75–0.99)** | **0.85 (0.74–0.98)** | **0.76 (0.60–0.96)** | **0.75 (0.59–0.94)** |
| ≥4 | **0.80 (0.63–0.99)** | **0.78 (0.61–0.99)** | **0.36 (0.24–0.55)** | **0.35 (0.23–0.53)** |
| Acute myocardial infarction | 0.86 (0.68–1.09) | 0.80 (0.60–1.06) | 0.93 (0.68–1.29) | 1.02 (0.70–1.47) |
| Ischaemic heart disease | 0.98 (0.82–1.17) | 0.93 (0.78–1.12) | 0.97 (0.70–1.36) | 0.96 (0.68–1.36) |
| Stroke | 0.96 (0.74–1.24) | 1.02 (0.74–1.39) | **0.65 (0.43–0.98)** | **0.61 (0.38–0.98)** |
| Diabetes | **0.48 (0.34–0.69)** | **0.49 (0.34–0.70)** | **0.49 (0.26–0.91)** | **0.46 (0.24–0.86)** |
| Chronic pulmonary disease | 0.94 (0.81–1.07) | 0.91 (0.79–1.06) | 0.91 (0.67–1.22) | 0.91 (0.67–1.25) |
| Renal disease | 0.72 (0.49–1.08) | 0.67 (0.44–1.03) | **0.55 (0.31–0.96)** | **0.49 (0.27–0.87)** |
| Dementia | **0.44 (0.29–0.67)** | **0.42 (0.27–0.65)** | 0.80 (0.49–1.31) | 0.85 (0.52–1.44) |
| Cancer | 1.01 (0.89–1.16) | 1.01 (0.88–1.15) | 0.95 (0.75–1.19) | 0.90 (0.71–1.14) |
| Peptic ulcer | **0.85 (0.74–0.99)** | **0.83 (0.72–0.97)** | 1.17 (0.87–1.57) | 1.15 (0.85–1.56) |
| Arrhythmias | 0.80 (0.65–0.99) | 0.83 (0.67–1.02) | 0.99 (0.75–1.30) | 0.96 (0.72–1.27) |
| Aged care residency | **0.69 (0.49–0.98)** | 0.71 (0.49–1.02) | 1.47 (0.84–2.57) | 1.43 (0.80–2.58) |
| Disability | 1.09 (0.91–1.30) | 1.11 (0.92–1.34) | 1.20 (0.85–1.68) | 1.33 (0.94–1.89) |
| Smoker | 1.08 (0.85–1.37) | 1.09 (0.86–1.39) | 0.97 (0.54–1.75) | 1.02 (0.56–1.86) |
| Private health insurance | 1.00 (0.90–1.11) | 1.02 (0.90–1.14) | 1.13 (0.91–1.41) | 1.12 (0.89–1.42) |
| Married | 0.98 (0.87–1.09) | 1.03 (0.93–1.15) | 0.87 (0.69–1.10) | 0.86 (0.68–1.06) |

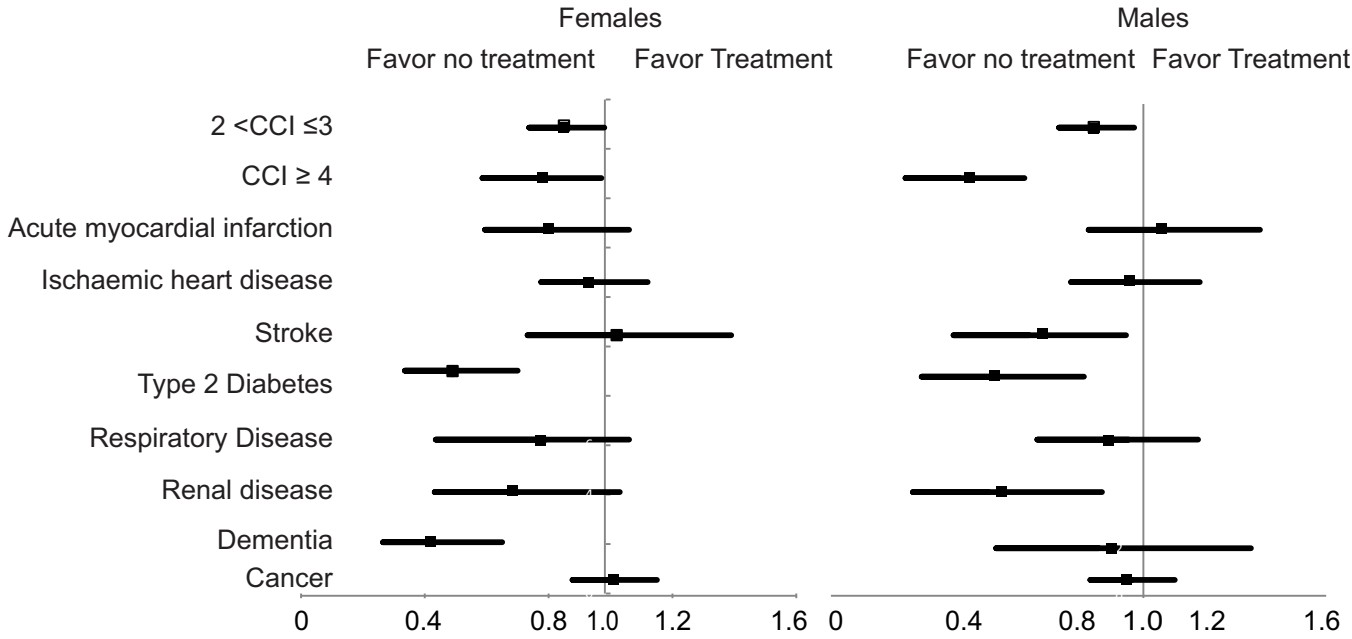

**Fig 2. Association between chronic conditions and treatment initiation in the high-risk group.**

## Estimation of the number of fractures prevented by treatment

In females, there were 1,149 recurrent fractures (369 hip) and 1,445 deaths during a median follow-up of 3 years (IQR: 1 to 5), yielding incidences of 44 any refracture (95% CI, 42 to 48), 14 hip refractures (95% CI, 13 to 16), and 51 deaths (95% CI, 48 to 53) per 1,000 person-years. Assuming a reduction of fracture risk by 30%, we estimated that osteoporosis treatment uptake could prevent 13 subsequent fractures (95% CI, 12 to 15) and 4 second hip fractures (95% CI, 3 to 5) per 1,000 person-years.

In males, there were 424 recurrent fractures (203 hip) and 1,460 deaths during a median follow-up of 2 years (IQR: 0.5 to 3), yielding incidences of 69 any refracture (95% CI, 62 to 76), 33 hip refractures (95%CI, 29 to 38), and 213 deaths (95% CI, 202 to 224) per 1,000 person-years. Treatment could have prevented 21 subsequent fractures (95% CI, 16 to 25) and 10 second hip fractures (95% CI, 7 to 13) per 1,000 person-years.

## Association of multimorbidity with subsequent fracture and mortality risk

Mild morbidity (CCI between 2 and 3) was associated with increased risk of subsequent fracture [HR, 1.18 (95% CI, 1.04 to 1.34) and 1.24 (95% CI, 1.06 to 1.44) for females and males, respectively] and increased risk of mortality [HR, 1.56 (95% CI, 1.26 to 1.92) and 1.53 (95% CI, 1.23 to 1.91), for females and males, respectively]. Severe multimorbidity (CCI ≥4) was also associated with increased risk of subsequent fracture [HR, 1.56 (95% CI, 1.26 to 1.92) and 1.53 (95% CI, 1.23 to 1.91) for females and males, respectively] and increased risk of mortality [HR, 5.16 (95% CI, 4.34 to 6.00) and 4.15 (95% CI, 3.68 to 4.69)].

## Sensitivity analyses

**Subset analysis of individuals with a 10-year Garvan Fracture Risk ≥40%.** This group included 3,323 females and 976 males, of whom only 9% of females and 3% of males received a DXA referral and 24% of females and 10% of males received osteoporosis treatment during a

median follow-up of 2.5 (IQR: 1 to 5) years. Predictors for DXA and treatment were similar to the main analysis.

In this group, treatment would have prevented 16 to 24 any refractures and 5 to 7 hip refractures per 1,000 person-years of follow-up.

**Sensitivity analysis regardless of 10-year Garvan Fracture Risk (hip and vertebral fractures and all fractures).** Of the total 25,280 participants (*n* = 16,191 females and *n* = 9,089 males), 3,479 (21%) females and 2,581 (28%) males sustained a hip or vertebral index fracture. Of these, 1,339 (38%) females and 372 (14%) males received prior osteoporosis treatment and were further excluded from this analysis.

Mild multimorbidity (CCI between 2 and 3) was significantly associated with lower odds for DXA in both sexes [adjusted OR; 0.71 (95% CI, 0.56 to 0.91) and 0.68 (95%CI, 0.52 to 0.90) for females and males, respectively] (S2 Table), and lower odds for treatment in females only [adjusted OR, 0.79 (95% CI, 0.62 to 0.99) and 1.08 (95% CI, 0.92 to 1.53) for females and males, respectively] (S3 Table). Severe multimorbidity (CCI ≥4) was associated with both lower odds for DXA [adjusted OR, 0.36 (0.19 to 0.71) and 0.34 (0.20 to 0.60), for females and males, respectively] (S2 Table) and treatment initiation [adjusted OR, 0.70 (95% CI, 0.48 to 1.00) and 0.46 (95% CI, 0.30 to 0.71) for females and males, respectively] for both sexes (S3 Table).

The association between comorbidities and DXA investigation and osteoporosis treatment was also observed when all participants were analysed together.

**Association between DXA and multimorbidity in all participants regardless whether they received osteoporosis treatment.** There were in total 2,527 (32%) DXA in females and 341 (13%) in males. The predictors for DXA were similar to the main analysis (S4 Table). Mild multimorbidity (CCI between 2 and 3) was associated with lower odds of DXA in both sexes [adjusted OR, 0.73 (95% CI, 0.64 to 0.84) and 0.46 (95% CI, 0.35 to 0.61) for females and males, respectively]. Similar findings were observed for severe multimorbidity (CCI ≥4) [adjusted OR, 0.51 (95% CI, 0.40 to 0.65) and 0.38 (95% CI, 0.25 to 0.57)].

## Discussion

We identified a group of individuals with an index fracture and a 10-year risk of any fracture ≥20% who were not currently being treated for osteoporosis, for whom further osteoporosis investigation and treatment would be warranted. In this group, <1/4 were investigated for osteoporosis and <1/3 received pharmacotherapy. A higher burden of comorbidity was associated with a lower likelihood of DXA referral and treatment. The association of multimorbidity with osteoporosis management was significantly greater in males than in females. This difference is partly explained by males' significantly higher mortality. Finally, we have estimated that treatment uptake could have prevented 13 to 21 osteoporotic fractures and 4 to 10 hip fractures per 1,000 person-years of follow-up. This estimate is based on fractures and deaths that actually occurred.

Our study found that multimorbidity was significantly associated with osteoporosis management gap. Among high-risk individuals, 17% females and 7% of males received only a DXA referral and 22% of females and 14% males received osteoporosis medication. A higher burden of multimorbidity was associated with up to a 57% lower probability of being investigated and up to 65% lower probability of being treated. A recent Danish study [16] reported a similar association of comorbidity and osteoporosis treatment. These findings indicate that the risk of fracture is either underestimated or underprioritised in people with multiple chronic conditions.

Type 2 diabetes was associated with a lower likelihood of both investigation and treatment in both sexes. It is possible that the higher BMI and implicitly higher bone density often

encountered in people with diabetes may contribute to the perception that they have at a lower fracture risk. Alternatively, the higher treatment burden among patients with diabetes may reduce patient and provider prioritization of fracture. Importantly, two recent meta-analyses have reported increased risk of fracture in both females and men with diabetes and, in particular, hip fracture [28,29]. Another recent meta-analysis demonstrated that diabetes did not affect the efficacy of osteoporosis medication [26–28,30]. Therefore, osteoporosis evaluation and medication initiation in patients with diabetes after an osteoporotic fracture should be considered.

Dementia was also associated with a significant lower likelihood of being investigated and treated, particularly in females. Numerous studies have reported an association between dementia and hip fracture [27–29,31], and a recent study presented evidence of the association of dementia with other types of fractures [28–30,32]. The osteoporosis treatment gap in people with dementia has also been previously reported, with only 5% of hip fracture patients with dementia receiving pharmacotherapy [29–31,33]. Furthermore, fracture prevention in people with dementia is uniquely challenging, requiring specific pathways to address the multiple shared risk factors [29–31,33].

This study confirms that osteoporosis management was negatively associated with male sex. Most males in the high-risk group remained undiagnosed and untreated following fracture. This is so even though following fracture, increases in subsequent fracture risk are equivalent in both sexes [34,35]. Importantly, this study also revealed that the sex disparity was greatest in the group with the highest burden of comorbidities. While this difference may be explained in part by males' higher competing mortality risk, our findings suggest that fracture risk is underappreciated or underprioritised in males.

This study's strengths include thorough documentation of chronic conditions and osteoporosis investigation and treatment in a large population-based cohort [22,23]. This allowed in-depth analysis of the relationship between multimorbidity and both fracture risk and the osteoporosis treatment gap. However, there are some limitations. The Sax Institute's 45 and Up Study cohort is relatively healthy and well educated and therefore may not be representative of the general population [4]. Thus, it is possible that the magnitude of association of multimorbidity with the outcomes may have been underestimated. Diagnosis miscoding is an inherent confounder in administrative health database studies. Self-reported comorbidities may be subjected to recall bias, and classification of comorbidities based on prescription would possibly underdiagnosed conditions without a specific pharmacological medication. In order to account for these limitations we have used multiple data sources for comorbidities identification, strategy that has been proven to be the best for comorbidities classification [36]. We used weight for the calculation of Garvan subsequent fracture estimates, which has approximately 10% lower predictive accuracy than the model with bone mineral density (BMD) [37]. Another possible limitation is that the Garvan Fracture Risk Calculator does not take into account the competing risk of mortality, which increases with multimorbidity. However, we estimated the number of fractures prevented by treatment based on the documented fractures and deaths that occurred in this high-risk population. We did not have information on calcium and vitamin D, which may be considered by some medical practitioners as fracture prevention therapy. We assumed that the efficacy of fracture-preventing medications is roughly equivalent in clinical trial participants and high-multimorbidity patients. This assumption requires further scrutiny. Our methods could not identify osteoporosis treatment initiated in hospital following fracture. However, the number of in hospital treatment initiations appears very small [16]. Moreover, treatment continued after discharge would have been captured. We have included all participants who received at least one osteoporosis prescription. We therefore cannot assess

adherence to prescribed treatment. Lastly, we could not ascertain in this study the number of participants with a clear contraindication for treatment.

In summary, this study found that multimorbidity at the time of fracture is significantly associated with a lower likelihood of being investigated and treated for osteoporosis. Diabetes, renal disease, and dementia were associated with lower rates of DXA referral and pharmacotherapy. The magnitude of association between chronic conditions and osteoporosis investigation and treatment was greater in males than females. Furthermore, we estimated that if treatment were equally efficacious in this high-risk, high-multimorbidity population as in clinical trial participants, 13 to 21 subsequent fractures and 4 to 10 hip fractures per 1,000 person-years of follow-up might have been prevented. Notably, these findings are based on a "relatively healthy" cohort, and thus the magnitude of association between multimorbidity and osteoporosis investigation and treatment may be underestimated in this study.

These findings suggest that fracture risk is either underestimated or fracture consequences are underappreciated in the context of multimorbidity. Individuals with multiple chronic conditions already have a high burden of illness, and low osteoporosis treatment uptake may accelerate their decline. Thus, the development of a more robust framework for fracture prevention in complex patients is warranted.

## Supporting information

**S1 STROBE Checklist. STROBE Checklist.**
(DOCX)

**S1 Table. Codes used for the identification of chronic conditions.**
(DOCX)

**S2 Table. Clinical risk factors associated with DXA investigation following index hip and vertebral fracture regardless of 10-year Garvan Fracture Risk estimate.**
(DOCX)

**S3 Table. Clinical risk factors associated with treatment investigation following index hip and vertebral fracture regardless of 10-year Garvan Fracture Risk estimate.**
(DOCX)

**S4 Table. Clinical risk factors associated with DXA investigation in all participants regardless of treatment initiation.**
(DOCX)

## Acknowledgments

This research was conducted using data collected through the Sax Institute's 45 and Up Study (www.saxinstitute.org.au). This study is managed by the Sax Institute in collaboration with major partner Cancer Council NSW; and partners: the Heart Foundation; NSW Ministry of Health; NSW Department of Communities and Justice; and Australian Red Cross Lifeblood. We thank the many thousands of people participating in the 45 and Up Study. We pay tribute to our colleague and collaborator, the late Dr Jian Sheng "Charles" Chen, who died before his work in establishing the initial research proposal, ethics approvals, and setting up the data linkages to answer important questions about osteoporotic fractures could be realized.

## Author Contributions

**Conceptualization:** Dana Bliuc, Kristine E. Ensrud.

**Data curation:** Dana Bliuc, Dunia Alarkawi.

**Formal analysis:** Dana Bliuc.

**Funding acquisition:** Dana Bliuc, Jacqueline R. Center.

**Investigation:** Dana Bliuc, Thach Tran, Weiwen Chen, Dunia Alarkawi, Dima A. Alajlouni, Fiona Blyth, Lyn March, Kristine E. Ensrud, Robert D. Blank, Jacqueline R. Center.

**Methodology:** Dana Bliuc, Thach Tran, Weiwen Chen, Dunia Alarkawi, Dima A. Alajlouni, Fiona Blyth, Lyn March, Kristine E. Ensrud, Robert D. Blank, Jacqueline R. Center.

**Project administration:** Dana Bliuc, Fiona Blyth, Lyn March.

**Resources:** Dana Bliuc, Fiona Blyth, Lyn March, Jacqueline R. Center.

**Software:** Dana Bliuc, Thach Tran, Weiwen Chen, Dunia Alarkawi, Dima A. Alajlouni, Fiona Blyth, Jacqueline R. Center.

**Supervision:** Kristine E. Ensrud, Robert D. Blank.

**Visualization:** Dana Bliuc, Dima A. Alajlouni.

**Writing – original draft:** Dana Bliuc.

**Writing – review & editing:** Dana Bliuc, Thach Tran, Weiwen Chen, Dunia Alarkawi, Dima A. Alajlouni, Fiona Blyth, Lyn March, Kristine E. Ensrud, Robert D. Blank, Jacqueline R. Center.

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
