## [Editor Report · Decision Letter 0]

27 Jun 2022

Dear Dr Bliuc, 

Thank you for submitting your manuscript entitled "The contribution of multimorbidity to under-diagnosis and under-treatment of osteoporosis in high-risk fracture patients: a prospective cohort study" for consideration by PLOS Medicine.

Your manuscript has now been evaluated by the PLOS Medicine editorial staff and I am writing to let you know that we would like to send your submission out for external peer review.

Please re-submit your manuscript within two working days, i.e. by Jun 29 2022 11:59PM.

Kind regards,

Callam Davidson

Associate Editor

PLOS Medicine

---

## [Decision Letter · Decision Letter 1]

8 Sep 2022

Dear Dr. Bliuc,

Thank you very much for submitting your manuscript "The contribution of multimorbidity to under-diagnosis and under-treatment of osteoporosis in high-risk fracture patients: a prospective cohort study" (PMEDICINE-D-22-02127R1) for consideration at PLOS Medicine. 

Your paper was evaluated by an associate editor and discussed among all the editors here. It was also discussed with an academic editor with relevant expertise, and sent to independent reviewers, including a statistical reviewer. The reviews are appended at the bottom of this email and any accompanying reviewer attachments can be seen via the link below:

[LINK]

In light of these reviews, I am afraid that we will not be able to accept the manuscript for publication in the journal in its current form, but we would like to consider a revised version that addresses the reviewers' and editors' comments. Obviously we cannot make any decision about publication until we have seen the revised manuscript and your response, and we plan to seek re-review by one or more of the reviewers. 

We hope to receive your revised manuscript by Sep 29 2022 11:59PM. Please email us (plosmedicine@plos.org) if you have any questions or concerns.

We look forward to receiving your revised manuscript. 

Sincerely,

Callam Davidson, 

PLOS Medicine

plosmedicine.org

Comments from the Academic Editor:

1/ The rationale may have been set in part by convenience based on the available dataset

2/ It is unclear how reliable the classification of patients with and without comorbidities is

3/ Reporting is lacking important information, especially in the results section where p values are presented alone, without the results.

Please update your title to ‘The association of multimorbidity and osteoporosis investigation and treatment in high-risk fracture patients in Australia: a cohort study’.

Please structure your abstract using the PLOS Medicine headings (Background, Methods and Findings, Conclusions).

Abstract Background: Provide the context of why the study is important. The final sentence should clearly state the study question.

Your study is observational and therefore causality cannot be inferred. Please remove language that implies causality, such as ‘impact’ and ‘contribution’. Refer to associations instead.

Abstract Methods and Findings:

* Please ensure that all numbers presented in the abstract are present and identical to numbers presented in the main manuscript text.

* Please include the years during which the study took place and length of follow up.

* Please quantify the main results (with 95% CIs and p values).

* Please include the important dependent variables that are adjusted for in the analyses.

The Data Availability Statement (DAS) requires revision. If the data are not freely available, please describe briefly the ethical, legal, or contractual restriction that prevents you from sharing it. If the data are available upon request, please note this and state the owner of the data set and contact information for data requests (web or email address). Note that a study author cannot be the contact person for the data.

Please do not use footnotes (such as those used in the Abstract). Relevant information should be included in the main text. 

Please update citations to be non-superscript, placed in square brackets, and preceding punctuation.

The terms gender and sex are not interchangeable (as discussed in https://www.who.int/health-topics/gender); please use the appropriate term.

Please provide a reference for the statement at line 198-199.

Please correct the typo at line 144. 

Please specify whether informed consent was written or oral.

Please define "lost to follow-up" as used in this study. Other reasons for exclusion should be defined.

Please remove the ‘Disclosures’ section from the end of the manuscript and ensure all relevant information is captured in the ‘Competing Interests’ section of the submission form. 

Please use et al only after listing the first six authors in your References.

Table 1: Please define Rx in the legend.

Please ensure that the study is reported according to the STROBE guideline, and include the completed STROBE checklist as Supporting Information. Please add the following statement, or similar, to the Methods: "This study is reported as per the Strengthening the Reporting of Observational Studies in Epidemiology (STROBE) guideline (S1 Checklist)."

Did your study have a prospective protocol or analysis plan? Please state this (either way) early in the Methods section.

Comments from the reviewers:

Reviewer #1: See attachment

Michael Dewey

Reviewer #2: This is an interesting and timely paper as the treatment gap in patients at high risk of fracture is receiving much attention. The potential role of comorbidities in the investigation and treatment of these patients is of interest. I would make the following comments:

a) I would suggest that the title is altered to say 'association' instead of 'contribution' as we can't ascribe causality.

b) I am a little unclear why patients with fractures, including hip and/or vertebral fractures, with Garvan risk scores <20 were not included in the analysis as many if not most guidelines would recommend strong consideration of investigation and/or treatment in such individuals. Can this be clarified? Do data for this group exist?

c) I am also a little unsure of the rational to look at DXA measurement ONLY in those who didn't subsequently get treatment - I can't see why this shouldn't be explored in all patients with perhaps the sub-group analyses as a sensitivity analysis.

Reviewer #3: This manuscript by Bliuc et al is an original prospective study assessing the impact of comorbidities on DXA referral and treatment initiation in a population cohort after an index fracture, in patients at high risk of refracture and not already receiving anti-osteoporotic drugs.

They show that overall in this population the rate of DXA referral is only 17% in women 7% in men, and treatment initiation is 22% in women and 14% in males. 

The higher the Charlson Comorbidity Index, the lower chance the patients are assessed for DXA or treated, in particular in men.

If these findings have been pointed out previously, there is little published evidence addressing specificly the interaction of various comorbidities on DXA screening and treatment initiation. This study by Bliuc et al clearly show that the the risk of fracture is either under-estimated or under prioritized in people with multiple chronic conditions. 

The reasons for that finding cannot be explained here, but this fact should be reflected and considered by the medical community, whatever the specialty of the reader is. The prevention of osteoporotic fractures is largely suboptimal, in particular in patients at higher risk. In other diseases, such as cardiovascular diseases for example, such a finding would be unaccepted. 

As an illustration, in Plos Medicine system, osteoporosis field of expertise is under Woman's health ! It should be under Endocrinology or/and Rheumatology!

It is robust and important for the medical community, way beyond bone health specialists.

Main remarks:

- The inclusion criteria was based on prior fracture (index fracture), Gravan Fracture risk without DXA > 20% , meaning that it is mainly on falls and weight that patients were selected. How does the CCI correlate with Gravan Fracture? Why was the Gravan risk used to select the patients since they are already at higher risk of fracture as they have a fracture history? Would the results be the same in patients with Gravan Fracture < 20 %? Would the rate of patients in each category (DXA / Tt initiation) be simila ? Could dementia and diabetes be significantly associated with falls?

- Many non bone health specialists physicians have in mind that Calcium and vitamin D are effective anti-osteoporotic drugs reducing fracture risk. Did the authors search for calcium and vitamin D alone in the cohort?

- Regarding competing risk of mortality, would it be possible to show the data for males and females in each CCI group?

- Discussion: could the authors speculate on the reasons for these findings? What would be the percentage of patients with a clear contra-indication of therapy?

Minor remark:

Table 1: Please indicate what Rx / NoRx stands for (treatment initiation?)

[LINK]

---

## [Decision Letter · Decision Letter 2]

4 Nov 2022

Dear Dr. Bliuc,

Thank you very much for re-submitting your manuscript "Multimorbidity is associated with lower chance of osteoporosis investigation and treatment in high-risk fracture patients in Australia: a prospective cohort study" (PMEDICINE-D-22-02127R2) for review by PLOS Medicine.

I have discussed the paper with my colleagues and the academic editor and it was also seen again by three reviewers. I am pleased to say that provided the remaining editorial and production issues are dealt with we are planning to accept the paper for publication in the journal.

[LINK]

We look forward to receiving the revised manuscript by Nov 11 2022 11:59PM.   

Sincerely,

Callam Davidson, 

Associate Editor 

PLOS Medicine

plosmedicine.org

Requests from Editors:

Your title must be nondeclarative – please revise to ‘The association between multimorbidity and osteoporosis investigation and treatment in high-risk fracture patients in Australia: a prospective cohort study’, or similar. 

Please relocate the ‘Limitation’ section of your abstract such that this limitation is described in the last sentence of the abstract Methods and Findings subsection (the abstract should only have three sections). 

Please combine bullets two and three in your Author Summary (‘High-risk patients often have multiple chronic conditions, and it is important to know to what degree these might alter…’).

Please change ‘women and men’ to ‘adults’ in your Author Summary.

Please quantify your key findings (with point estimates and 95% CI) in your Author Summary.

Thank you for completing the STROBE checklist. As the final published version will not contain line numbering, please update the checklist to refer to section headings and paragraph numbers instead.

Line 232: Please include references to support this statement. 

Lines 330 and 331: Please do not report P<0.000#, please report as P<0.001. 

Please refer to males and females rather than men and women to reflect the fact that sex rather than gender was determined. Please check throughout the text as well as in Figures. 

As your study was observational, please avoid language that implies causality and refer instead to associations (examples at lines 456 and 485).

Please remove the funding section from the end of the main manuscript and ensure all relevant information is captured in the Financial Disclosure in the submission form.

Please remove the Author roles section (this is captured as metadata during the submission process).

Table 1: Given that the majority of metrics are presented as N (%), it may be more simple to remove footnotes 2/3, update the column header (‘%/SD’) and instead state ‘All figures are N (%) unless otherwise stated’. Then only one footnote is needed for Age.

Comments from Reviewers:

Reviewer #1: The authors have addressed my points

Michael Dewey

Reviewer #2: The authors have addressed the reviewers' comments and I think the manuscript is much improved.

Reviewer #3: Dear authors,

all the comments have been answered and I have no further comment or question. I'm happy with the revised manuscript and would like to congratulate the authors for this important work.

I hope that the dissemination of these findings will help reducing the gap of diagnosis and treatment of osteoporotic patients in general.

Best regards

[LINK]

---

## [Editor Report · Decision Letter 3]

18 Nov 2022

Dear Dr Bliuc, 

On behalf of my colleagues and the Academic Editor, Professor Christelle Nguyen, I am pleased to inform you that we have agreed to publish your manuscript "The association between multimorbidity and osteoporosis investigation and treatment in high-risk fracture patients in Australia: a prospective cohort studyd" (PMEDICINE-D-22-02127R3) in PLOS Medicine.

Please also make the following editorial changes:

* There is a typo in the title ('studyd')

* Line 60: 'The cohort is relatively heathy, therefore...' - this sentence needs to be relocated to just after line 56 (it should be the last sentence of the Abstract Methods and Findings, rather than the Abstract Conclusions).

* Line 78: Please update this bullet to: 'We conducted a prospective study including over 10,000 adults aged 50+ with an osteoporotic fracture and high-risk of future fracture to investigate the association between multimorbidity and osteoporosis investigation and treatment'.

PRESS

Sincerely, 

Callam Davidson 

Associate Editor 

PLOS Medicine